# Overexpression of *MdCPK1a* gene, a calcium dependent protein kinase in apple, increase tobacco cold tolerance via scavenging ROS accumulation

Hui Dong[1], Chao Wu[1], Changguo Luo[2], Menghan Wei[1], Shenchun Qu[1], Sanhong Wang[1]*

**1** College of Horticulture, Nanjing Agricultural University, Nanjing, China, **2** Guizhou Fruit Institute, Guizhou Academy of Agricultural Science, Guiyang, China

* wsh3xg@hotmail.com

**Data Availability Statement:** All relevant data are within the manuscript and its Supporting Information files.

**Funding:** This work was supported by the National Natural Science Foundation of China (31872076,

## Abstract

Calcium-dependent protein kinases (CDPKs) are important calcium receptors, which play a crucial part in the process of sensing and decoding intracellular calcium signals during plant development and adaptation to various environmental stresses. In this study, a CDPK gene *MdCPK1a*, was isolated from apple (*Malus×domestica*) which contains 1701bp nucleotide and encodes a protein of 566 amino acid residues, and contains the conserved domain of CDPKs. The transient expression and western blot experiment showed that MdCPK1a protein was localized in the nucleus and cell plasma membrane. Ectopic expression of *MdCPK1a* in *Nicotiana benthamiana* increased the resistance of the tobacco plants to salt and cold stresses. The mechanism of MdCPK1a regulating cold resistance was further investigated. The overexpressed *MdCPK1a* tobacco plants had higher survival rates and longer root length than wild type (WT) plants under cold stress, and the electrolyte leakages (EL), the content of malondialdehyde (MDA) and reactive oxygen species (ROS) were lower, and accordingly, antioxidant enzyme activities, such as superoxide dismutase (SOD), peroxidase (POD) and catalase (CAT) were higher, suggesting the transgenic plants suffered less chilling injury than WT plants. Moreover, the transcript levels of ROS-scavenging and stress-related genes were higher in the transgenic plants than those in WT plants whether under normal conditions or cold stress. The above results suggest that the improvement of cold tolerance in *MdCPK1a*-overexpressed plants was due to scavenging ROS accumulation and modulating the expression of stress-related genes.

## Introduction

Abiotic stresses, such as drought, high salinity, cold or submergence, are serious threats to crop productivity. Plants have evolved fine signaling strategies enabling them to overcome these stresses and other harmful conditions. Among the strategies adopted by plants, calcium signals are important regulators in many crucial and sophisticated cellular processes [1].

31560551). There was no additional external funding received for this study.

**Competing interests:** The authors have declared that no competing interests exist.

When plants are subjected to various stresses, they rapidly release calcium ions ($Ca^{2+}$) from storage compartments (such as vacuole, endoplasmic reticulum) into the cytosol. Transient increases of free $Ca^{2+}$ in cytosolic are perceived and decoded through different $Ca^{2+}$ sensors and $Ca^{2+}$ binding proteins, such as calcium-dependent protein kinases (CDPKs), calmodulin-like proteins, calmodulins and calcineurin B-like proteins. CDPKs distinguished from other calcium-sensing proteins, as they not only can decode and translate the increase of $Ca^{2+}$ concentration into improvement of protein kinase activity but also can activate downstream effectors [2].

CDPKs exist in protists, oomycetes, green algae and plants, but not in animals [3]. Genome-wide analysis of different plant species showed that they are encoded by a large multigene family. For example, *Oryza sativa*, *Zea mays*, *Malus domestic*a, *Populus trichocarpa*, and *Arabidopsis thaliana* were identified 31, 35, 37, 30 and 34 *CDPK* genes in their genomes, respectively [4–11]. CDPKs have a conserved modular structure including a variable N-terminal domain, a kinase domain, an auto-inhibitory domain or junction domain and a regulatory domain or CaM-like domain, which canonically contains four EF-hands [12, 13]. In the absence or low concentration of cytoplasmic $Ca^{2+}$, auto-inhibitory domain blocks the kinase domain and inhibits its activity [14, 15]. When plants perceive stimuli, an immediate increase of the concentration of $Ca^{2+}$ in plant intracellular promotes $Ca^{2+}$ binding to EF-hand motifs, which will induce molecular conformation changes and activate enzyme activities, leading to phosphorylation of the targeted substrates as well as CDPK autophosphorylations [16–18]. The phosphorylated proteins probably participate in plant defense reactions, ethylene synthesis, cytoskeleton organization, carbon and nitrogen metabolism, and stress responses [5, 17, 19–22]. Knowledge about CDPK functions and mechanisms of the responses to environmental stress is increasing. Substantial experimental evidences indicate CDPKs play important roles in response to abiotic/biotic stress. For example, *Arabidopsis CPK28* acts as a positive regulator in response to osmotic stress [23]. *OsCPK9* in rice plays a positive role in drought, osmotic, and dehydration stress responses [24]. Overexpressing of *OsCPK4*, *OsCPK12* in rice exhibited increased salt/drought stress tolerance and rice blast disease resistance [25–27]. *CaCDPK15* in pepper (*Capsicum annum*) positively regulates response to *Ralstonia solanacearum* [28]. In Arabidopsis, overexpression of *SiCDPK24* enhanced drought tolerance [29]. *OsCDPK1* positively regulates salt and drought tolerance in rice [30], meanwhile it acts as a positive regulator of *OsPR10a* participating in the defense signaling pathway [31]. Conversely, some *CDPKs* are negative regulators of stress response because transgenic plants overexpressing them are more sensitive to abiotic/biotic stresses. *Arabidopsis thaliana cpk23* mutant increased endurance to drought and salt stresses, while *AtCPK23* overexpressing plants reduced the resistance to drought and salt stresses [32]. Overexpression of *ZmCPK1* in maize mesophyll protoplasts suppressed the expression of the cold-induced marker gene *Zmerf3*, and ectopic expression of *ZmCPK1* in *Arabidopsis* reduces plants adaption to the cold tolerance, suggesting *ZmCPK1* act as a negative regulator of cold stress signalling in maize [33]. The *Arabidopsis CPK28* plays as a negative regulator of immune signaling that continually buffers immune signaling by controlling the turnover of BIK1, an important convergent substrate of multiple pattern recognition receptor (PRR) complexes [34]. Thus, *CDPKs* are implicated in both positive and negative regulation of plant abiotic/biotic stress adaptation.

However, the research on function of CDPKs in apple has been rarely reported. This study focused on the function of *MdCPK1a*, a CDPK gene from *M. domestica*, in response to abiotic stresses. *MdCPK1a*-overexpressed *N.benthamiana* plants were investigated to different abiotic stress conditions. Experimental results showed that overexpression of *MdCPK1a* in *N. benthamiana* confers it resistance to salt and cold stresses. Furthermore, the mechanism of enhancement of cold tolerance in the transgenic plants was disclosed in this research.

## Materials and methods

### Cloning, sequencing and phylogenetic analysis of *MdCPK1a*

The fourth and fifth young leaves were taken from the annual branches of the *Malus domestica* cv.'Jonathan' growing in the greenhouse. Total RNA was extracted by using CTAB method. [35]. Based on the released sequence of *MdCPK1a* (*MDP0000153100*) from Phytozome (https://phytozome. jgi.doe.gov/pz/portal.html), a pair of primers GSP1 was designed for gene amplification by RT-PCR (S1 Table). The PCR amplification was performed in a total 50 μL reaction volume containing 300 ng cDNA, 1×TransStart Fast Pfu buffer, 0.25 mM dNTPs, 0.4 mM of each primer and 2.5 units of TransStart Fast Pfu DNA polymerase. PCR conditions were set as follows: initial denaturation at 95˚C for 2 min; 40 cycles of 95˚C for 20 s, 55˚C for 20 s, and 72˚C for 60 s, and followed by a final extension at 72˚C for 5 min. The construction of PCR products ligation with *pMD19*-T vector were named *pMD19T-MdCPK1a*, and sequenced by Invitrogen (Shanghai, China).

The domain was identified through PROSITE and Smart™ databases (http://smart.embl-heidelberg.de/); Molecular weight and theoretical isoelectric point (pI) were calculated by ExPASy software (http://www.expasy.org/); The position of S-Palmitoylation and N-Myristoy-lation were predicted using the online tool GPS-Lipid (http://lipid.biocuckoo.org/presult.php) [36–38]. The homologous proteins were searched by BLASTp program (http://www.ncbi.nlm.nih.gov/) using the deduced amino acid sequence of *MdCPK1a*. Multialignment was performed by DNAMAN software (http://www.lynnon.com/). A phylogenetic tree was built through the neighbor-joining (NJ) method under the MEGA 6.0 program with Poisson-corrected distances with 500 bootstrap replicates.

### Subcellular localization analysis

The coding sequence of *MdCPK1a* with termination codon removal was amplified from *pMD19T-MdCPK1a* using primer GSP3 (S1 Table). Amplification products were digested with *Xba* I and *Bam*H I, and cloned into the downstream of *CaMV 35S* promoter in pCAM-BIA1300 vector resulting in *MdCPK1a* C-terminal in-frame fusion with GFP gene to form a plasmid *35S::MdCPK1a-GFP*. The *35S::MdCPK1a-GFP* and *35S::GFP* (control) constructs were transiently transformed into *N. benthamiana* leaves described by Sheludko [39]. GFP fluorescence was imaged under a laser confocal fluorescence microscopy (Zeiss TCS SP8) with an excitation wavelength of 488 nm and a 505–530 nm bandpass filter.

### Protein extraction and western blot

Tobacco leaves that transiently expressed *35S::MdCPK1a-GFP* and *35S::GFP* (control) were homogenized in liquid nitrogen. The nuclear proteins, cytoplasmic proteins and plasma membrane(PM) proteins were extracted with the Plant Nuclear, Cytoplasmic and Membrane Proteins Extraction Kit (BestBio, Shanghai, China) from plant tissues, respectively. Total proteins were extracted with extraction buffer (50 mM Tris-HCl, pH 7.5, 150 mM NaCl, 1 mM EDTA, 0.1%SDS, 1%Triton X-100), and followed by centrifugation at 12000 rpm for 15min, and the supernatants were collected.

Following standardization of protein concentrations using BCA Protein Assay kit (BestBio, Shanghai, China), Equal amounts of protein were employed in 10% SDS-PAGE and transferred to the NC membrane. After blocking with 5% skimmed milk powder in PBST (0.5% Tween in PBS) at room temperature for 2 h, the membrane was incubated with Anti-GFP rabbit polyclonal antibody (Sangon Biotech, Shanghai, China) at 4˚C overnight. After this, the membrane was rinsed three times with PBST for 5 min and then incubated with the HRP-

conjugated Goat Anti-Rabbit IgG (Sangon Biotech, Shanghai, China) for 1 h. Subsequently, the membrane was washed with PBST, visualized by enhanced chemiluminescence and then detected in the Tanon 2500 chemiluminescence imaging system (Shanghai, China).

## Overexpression of *MdCPK1a* in N.benthamiana

The full length ORF of *MdCPK1a* flanking *Bam*H I and *Sac* I at 5' and 3' respectively was amplified by the primers GSP2 (S1 Table). The PCR products were double-digested with *Bam*H I and *Sac* I, then ligated into the pYH455 vector at downstream of *CaMV35S* promoter (S2 Fig), generating a plasmid *pYH455-MdCPK1a*. Subsequently, it was transferred into EHA105. Tobacco transformation was conducted using leaf disk method [40]. Transgenic tobacco plants were selected on MS medium supplement 50 mg·L$^{-1}$ kanamycin. The kanamycin-resistance plants were further confirmed by PCR and RT-PCR respectively with the control of non-transformed tobacco plants cultured on MS medium.

## Abiotic tolerance analysis of the transgenic tobacco plants

Three independent lines (A2, A4 and A36) and wild type (WT) plants were used to analyze abiotic tolerance. After being surface disinfected, the seeds of A2, A4, A36 and WT were sown on MS medium (for transgenic lines MS medium supplemented 50 mg·L$^{-1}$ kanamycin). *N. benthamiana* were cultured under long day conditions 16 h light at 23–25˚C and 8 h dark at 18–20˚C.

To assess cold resistance, we grew the seedlings under cold stress (4˚C) for 10 d after seeds germinating on MS medium and measured the root length after cold treatment. Meanwhile, four-week-old plants growing in medium were stressed at 4˚C for 10 d, and then the survival rates were calculated after recovering at 25˚C for 14 d according to the number of green plants.

For salt or drought tolerance assays, 4-week-old plants were transplanted into soil with sufficient water under a normal environmental chamber at 25˚C for 14 d. They were then watered with 200 mM NaCl solution in soil for salt stress analysis, or cultured without irrigation for 25 d, and then recovered by re-watering for 10 d for drought stress analysis. The biomass and phenotype were investigated after the treatments.

## Physiological measurements and histochemical staining

Sixty-day-old plants treated at 4˚C for 48 h were used as material. Malondialdehyde (MDA) contents were measured using the thiobarbituric acid (TBA)-based colorimetric method [41]. Leaf samples (0.5 g) were homogenized in 2 mL 20% trichloroacetic acid with the aid of some sand, and then the homogenate was centrifuged at 16,000 g for 20 min at 4˚C. The supernatant (1 mL) was mixed with equal volume of 0.5% (w/v) TBA. The mixture was heated at 95˚C for 30 min and then quickly cooled in an ice bath. After centrifugation at 10,000 g for 10 min, absorbancy was measured at 532 nm corrected for nonspecific turbidity by subtracting the absorbancy at 600 nm. The MDA content was calculated using its molar extinction coefficient (155 mM$^{-1}$ cm$^{-1}$), and the value was expressed as μmol MDA· mg$^{-1}$ fresh weight (FW). Electrolyte leakages (EL) were detected using the protocol according to [42]. The leaf segments from at least three plants of each line were placed in deionized water for 2 h at 25˚C. Total electrolyte content was measured after autoclaving the leaf segments for 15 min and taken as 100% leakage. Activities of catalase (CAT), peroxidase (POD), and superoxide dismutase (SOD) were analyzed according to [43]. Leaf samples (0.2 g) were homogenized in liquid nitrogen adding 2 mL precooled 50 mM pH 7.8 phosphate buffer (containing 0.1 mM EDTA and 1% PVP) and ground into homogenate in an ice bath. Add extraction medium to rinse the mortar for 2–3

times and make the final volume 8.0 mL. The supernatant was centrifuged at 12 000 × g for 15 min at 4˚C and stored in the ice bath for the detection of SOD, POD, CAT activities. The accumulation of $H_2O_2$ and $O_2^-$ was tested by histochemical staining with nitroblue tetrazolium (NBT) and 3, 3'-diaminobenzidine (DAB) respectively. Leaves were incubated in the NBT solution (0.1 mg·mL$^{-1}$) and DAB solution (1.0 mg·mL$^{-1}$, pH 3.8) for 24 h at 25˚C in the dark. Then, the leaves were soaked in 95% ethanol overnight to remove the chlorophyll [44, 45]. DAB/NBT-stained leaves were scanned, and the pixel intensity of the DAB/NBT stain was quantified using Adobe PHOTOSHOP CS4 software.

## Quantitative RT-PCR analysis of gene expression in transgenic plants

The expression level of stress-related genes was monitored by quantitative RT-PCR (qPCR) on an ABI7300 Detection System using SYBR® Premix ExTaq™ qRT-PCR kits (TaKaRa, Dalian, China). Gene-specific primers were designed by Primer 5.0 (S1 Table). PCR mixtures contained 10.0 μL of 2×SYBR Premix, 1.0 μL of cDNA template, 200 nM of each primer, then added ddH$_2$O up to a total volume of 20.0 μL. PCR reaction was performed as follows: denaturation at 94˚C for 3 min followed by 40 cycles at 94˚C for 20 s, 60˚C for 20 s, and 72˚C for 40 s. After that, melting curves were determined as follows: 95˚C for 15 s, 60˚C for 1 min, and 95˚C for 15 s. qPCR was performed three independent biological repeats for each sample and three technical repeats for each reaction. Expression values were normalized with *NtTubulin* gene (Accession No: EF051136). The relative expression of a gene was calculated by using $2^{-\Delta\Delta Ct}$ method [$\Delta\Delta Ct = (Ct_{target\ gene} - Ct_{tubulin\ gene})$ treatment$-(Ct_{target\ gene} - Ct_{tubulin\ gene})$ control].

## Statistical analysis

Every experiment was repeated three times, and the value was got from an average from three independent replicates and shown with error bar representing with standard error (SE). All statistical analyses were performed using SPSS software and based on Duncan's multiple range tests, statistical differences were compared and p values <0.05 or <0.01 were used as the thresholds for significant or extremely significant differences, respectively.

## Results

### Cloning and bioinformatics analysis

The full length open reading frame (ORF) of *MdCPK1a* was isolated from apple, which consisted of 1701 nucleotides encoding a 566-amino acid polypeptide with the predicted molecular weight 62.86 kDa and the isoelectric point 5.16. *MdCPK1a* protein possesses the characteristics as other plant CDPKs: an N-terminal variable domain (107aa) preceding a Ser/Thr protein kinase catalytic domain (259 aa), a junction domain (42 aa), a CaM-like domain containing four EF hand $Ca^{2+}$-binding motifs (142 aa) and a C-terminal variable domain (16 aa). A possible ATP-binding site and active site in the N-terminal region and 15 invariant amino acid residues for eukaryotic Ser/Thr protein kinase in the N-terminal of kinase domain were shown in Fig 1. The putative post-translational modifications of *MdCPK1a* protein was predicted by the software GPS-Lipid, showing that there are one myristoylation (Gly at the 2$^{nd}$ residue from the N-terminus) and two palmitoylation (Cys at the 5$^{th}$ and 136$^{th}$ residues from the N-terminus) in the protein (Fig 1).

Multiple sequence alignments showed the deduced amino acid sequence of *MdCPK1a* with 72.73% similarity to *OsCDPK7* (BAB16888), 76% to *AtCPK1* (NP_196107), and 71.75% to *ZmCPK1* (BAA12338). The phylogenetic relationships between *MdCPK1a* and several stress-related CDPKs are presented in S1 Fig. The selected CDPK proteins were clustered into three

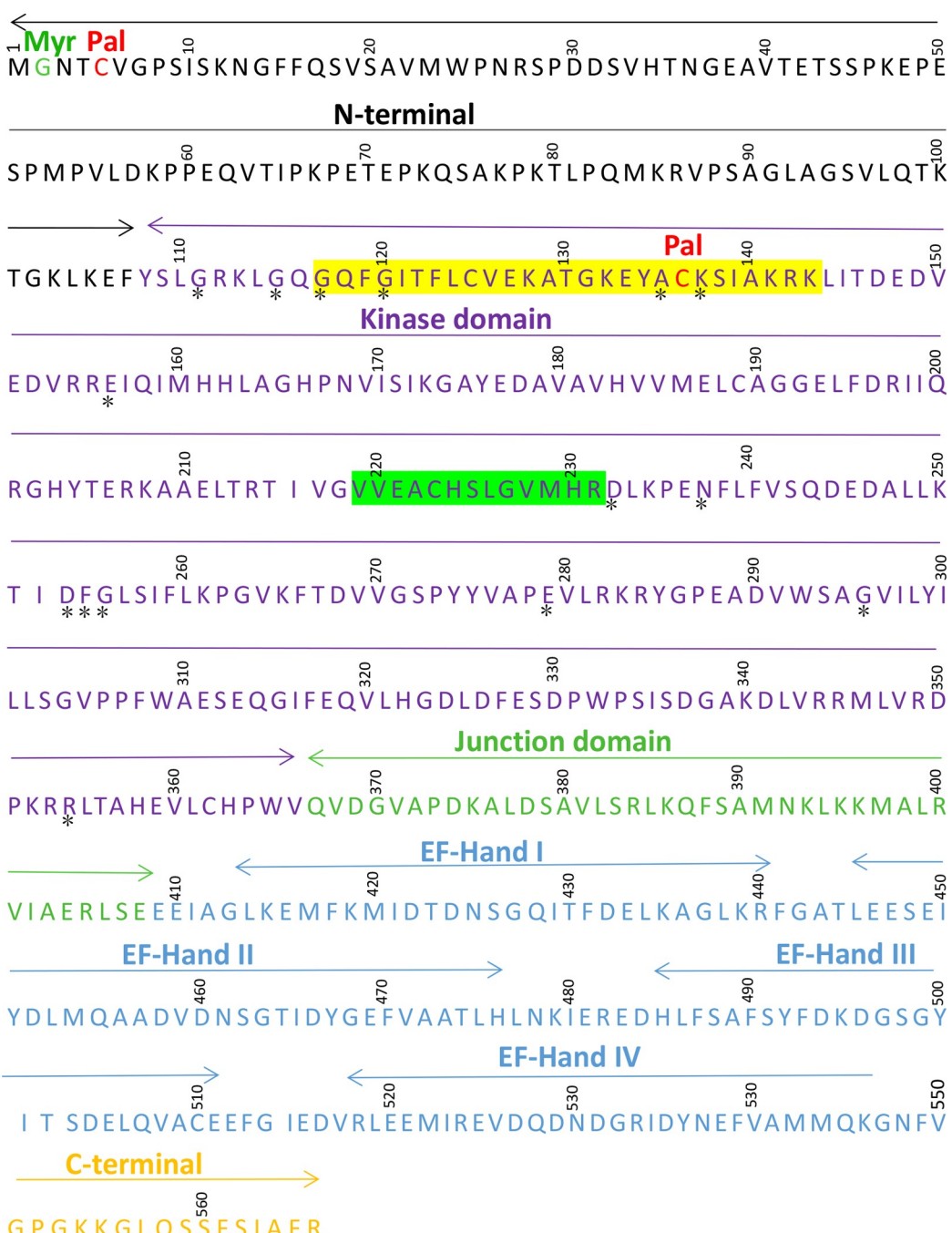

**Fig 1. Protein sequence analysis of MdCPK1a.** Kinase domains, Junction domain, and EF hand loops of CaM-LD domain of CDPK are marked. The 15 invariant amino acid residues for eukaryotic Ser/Thr protein kinase are indicated by asterisks. Protein kinase ATP-binding site and active site are highlighted in dark orange and green, respectively. The positions of predicted S-Palmitoylation (Pal) and N-myristoylation (Myr) are indicated in the diagram.

subgroups including I, II, III. *MdCPK1a*, along with *OsCDPK7*, *AtCPK1*, and *ZmCPK1* which were reported to regulate abiotic and biotic stress tolerances [33, 46, 47], belongs to the subgroups I, which hints that *MdCPK1a* may participate in stress responses in apple.

## Subcellular localization

The subcellular location of a protein determines or is closely correlated with its function. To investigate the subcellular location of *MdCPK1a* protein, we cloned the full-length ORF sequence of *MdCPK1a* into pCAMBIA1300 vector under *CaMV35S* promoter, constructing an in-frame fusion protein plasmid *35S::MdCPK1a-GFP* (Fig 2A). The construct was transformed to *N. benthamiana* leaves by agro-infiltration for transient expression analysis. The subcellular location of *MdCPK1a-GFP* was detected by laser scanning confocal microscopy, with the leaves transiently transformed *35S::GFP* as the control. The tobacco cells expressing the *35S::MdCPK1a-GFP* emitted fluorescence both in nucleus and plasma membrane, whereas in expressing the *35S::GFP* tobacco cells, the fluorescence filled the entire cytoplasm, plasma membrane and nucleus (Fig 2B). We further verified the subcellular localization by western blot by immunoblotting with anti-GFP antibody. MdCPK1a-GFP was detected exclusively in the fractions of plasma membrane and cell nucleus but not in the fraction of cytosol (Fig 2C). These results indicated that *MdCPK1a* protein was localized to the nucleus and cell plasma membrane.

## Overexpression of *MdCPK1a* gene in tobacco

The overexpression construct of *MdCPK1a* (*pYH455-MdCPK1a*) was introduced into *N. benthamiana* by *A. tumefaciens*-mediated transformation. Ten transgenic lines were obtained and further identified by PCR using gene-specific primers (GSP1). Six lines were randomly selected for gene transcription analysis. The result showed *MdCPK1a* was expressed constitutively in these lines, among which three independent lines (A4, A36, and A2) were used for analysis of the resistance to abiotic stresses. The levels of *MdCPK1a* mRNA in the three transgenic lines were quantified by qPCR. *MdCPK1a* mRNA displayed the highest level in A4 and the lowest level in A2 (S2 Fig).

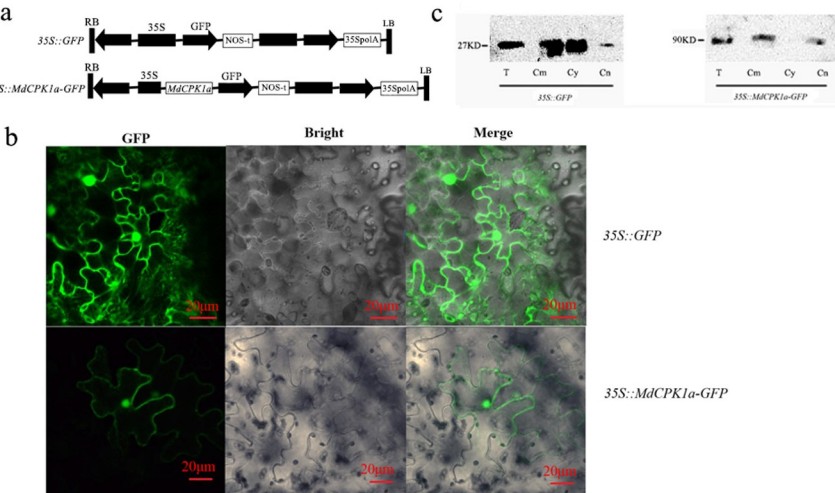

**Fig 2. Subcellular location of MdCPK1a.** (a) Schematic representations of the vector constructs of *35S::GFP* and *35S:: MdCPK1a-GFP*. (b) Subcellular localization of MdCPK1a-GFP fusion protein was conducted by transient expression experiment in *N. benthamiana* cells. Images were taken by using Leica confocal microscopy at 72 hours post agroinfiltration (GFP: fluorescence, green; Bright: visible light image; Merge: merged images of above two images). Bars = 20 μm. (c) Subcellular localization of MdCPK1a-GFP fusion protein was detected by western blot. Total protein extract (T), cell membrane fraction (Cm), cytosolic fraction (Cy), and cell nucleus fraction (Cn) isolated from *35S:: GFP*-expressing (left) *35S::MdCPK1a-GFP*-expressing (right) tobacco cells were immunoblotted with anti-GFP antibody.

## Stress tolerance of *MdCPK1a*-overexpressed *N. benthamiana* plants

For salt stress analysis, 6-week-old of WT and $T_2$ plants of A4, A36 and A2 were irrigated with 200 mM NaCl solution in the soil once a week. After suffering from salt stress for 25 d, WT plant leaves turn yellow, the *MdCPK1a*-overexpressed (*MdCPK1a-OX*) transgenic tobacco plants grew better and more vigorously compared with WT,(Fig 3A). The dry weight of shoots and roots of the transgenic lines, except A2, was significant higher than that of WT (Fig 3B), which suggests that *MdCPK1a-OX* tobacco plants increased the tolerance to salt stress. However, the similar symptoms between WT and the transgenic lines were observed under drought stress. The transgenic plants and WT displayed slow-growing, rolled and wilted leaves without irrigation for 25 d and showed no significant difference after re-watering for 10 d (Fig 4), suggesting that there were no obvious differences of drought resistance between WT and the transgenic lines.

The cold tolerance of seedlings of WT and the transgenic plants was monitored on MS medium at 4˚C. Both of them showed severe growth inhibition, however, the root length of WT was significantly shorter than that of the transgenic plants (Fig 5A and 5C). Meanwhile, we also analyzed the cold resistance of them at 4 weeks old. They were stressed in 4˚C for 10 d then recovered in normal conditions (25˚C) for 14 d. WT plants suffered from chilling injury more severely than A4, A36, and A2. Only 17% of WT plants survived after cold treatment, while 60–96% of the transgenic lines survived (Fig 5B and 5D). Physiological analysis showed that the transgenic plants had lower MDA content and less electrolyte leakage (EL) than WT plants under cold stress (Fig 5E and 5F), indicating that the transgenic plants were less injured compared with WT plants.

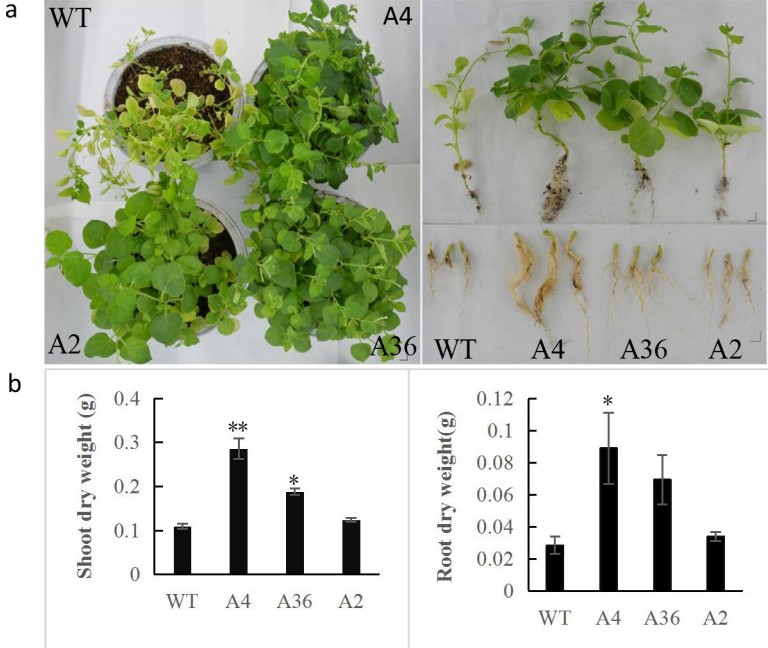

**Fig 3. Phenotype and stress tolerance of the *MdCPK1a-OX* plants under salt stress.** (a) Photographs of wild-type and *MdCPK1a-OX* plants (line A4, A36, and A2) under 200 mM NaCl. The left shows the phenotype of aerial part of tobacco plants, the right shows the phenotype of shoot and root. (b) Shoot and root dry weight of *MdCPK1a*-OX plants after the salt-stress treatment. Error bars indicate the standard error of the mean (SEM) of three independent experiments. Significant differences between the WT and transgenic plants are indicated by asterisks (*p< 0.05, **p < 0.01).

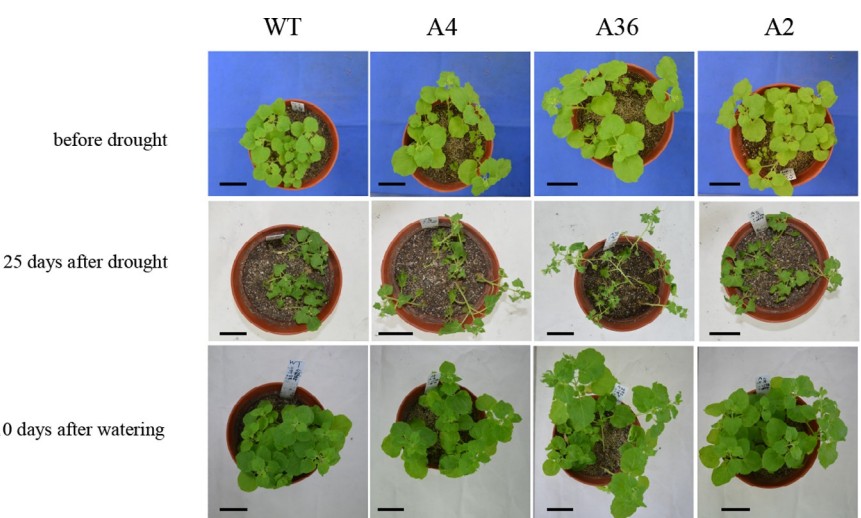

**Fig 4. Photographs of wild-type and *MdCPK1a*-OX plants under drought stress.** 4-week-old seedlings were transplanted into soil with sufficient water in the chamber at 25˚C for 14 d. They were cultivated for 25 d without watering for draught stress, and then re-watered for 10 d for recovery. Photographs of representative plants of WT and *MdCPK1a*-OX plants (A4, A36, and A2) were taken before and after the treatment of drought stress, and 10 days after re-watering, respectively. Twenty plants of each line were used for the experiments. Bars in picture = 10 cm.

## Analysis of ROS levels and antioxidant enzyme activities in transgenic *N. benthamiana*

Under abiotic stresses, reactive oxygen species (ROS) such as hydroxyl radical (·HO), superoxide radical ($O_2^{-}$) or hydrogen peroxide ($H_2O_2$), are excessively accumulated in plants, which

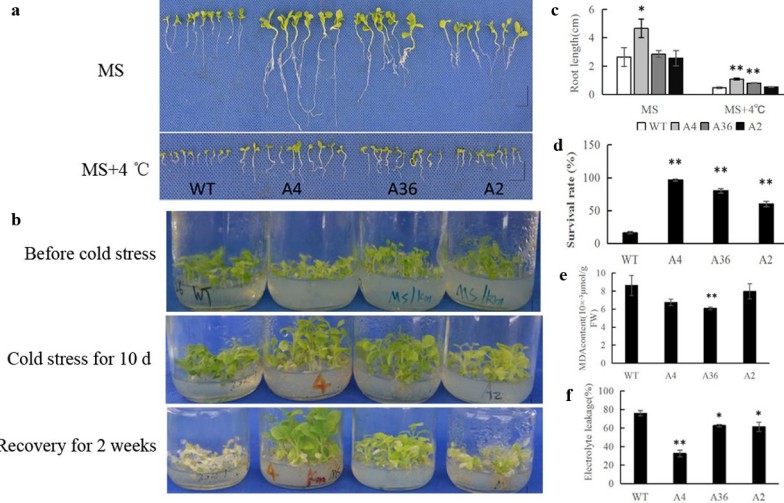

**Fig 5. Overexpression of *MdCPK1a* enhances cold tolerance in transgenic tobacco.** (a) Phenotype of seedlings of WT and *MdCPK1a*-OX plants (A4, A36 and A2) at the normal growth temperature and low temperature (4˚C) for 10 d on MS medium after germinated; (b) Responses to cold stress of 4-week-old WT and *MdCPK1a*-OX plants; (c) Root lengths of seedlings after germination for 10 d on MS medium at 4˚C; (d) Survival rates of WT and *MdCPK1a*-OX plants. Values are the mean ± SE. Thirty plants of each line were used for statistics; (e) Detection of MDA content in WT and *MdCPK1a*-OX plants. FW means Fresh weight. Data represent the means ± SE of at least three replicates; (f) Electrolyte leakage in WT and *MdCPK1a*-OX tobacco plants; Significant differences between the WT and transgenic plants are indicated by asterisks (*p< 0.05, **p < 0.01).

act as important signal molecules and also are toxic by-products leading to oxidative damage [48]. To reduce the damage of excessive production of ROS, plants have developed a scavenging mechanism allowed them to overcome ROS toxicity. To know whether *MdCPK1a* regulates ROS levels in cold response, we compared with the ROS levels in the overexpressing tobacco lines and WT plants after suffering cold stress. NBT and DAB staining were used to detect the accumulation of $O_2^{\cdot-}$ or $H_2O_2$ in leaves, respectively. Before the cold treatment, the leaves had similar dyeing degree in the transgenic plants and WT, indicating $O_2^{\cdot-}$ or $H_2O_2$ accumulation was similar in both plants. However, lower dyeing degree was detected in the transgenic plants whether by NBT (Fig 6A and 6C) or DAB (Fig 6B and 6D) staining under cold stress, suggesting less ROS accumulated in the transgenic plants than that in WT. Furthermore, compared with WT, the enzyme activities of CAT, POD and SOD of the transgenic lines were higher before treatment and significantly higher after 48h cold treatment. POD activity was still significantly higher in the transgenic plants while there was no significant difference in CAT and SOD activities after 72 h cold treatment (Fig 7).

## Expression analysis of the stress related genes in transgenic *N. benthamiana*

The transcriptional levels of stress-responsive and ROS-related genes (*NtSOD*, *NtGPX*, *NtCAT*, and one ROS-producing NADPH oxidase gene, *NtrbohD*) were analyzed by qPCR in

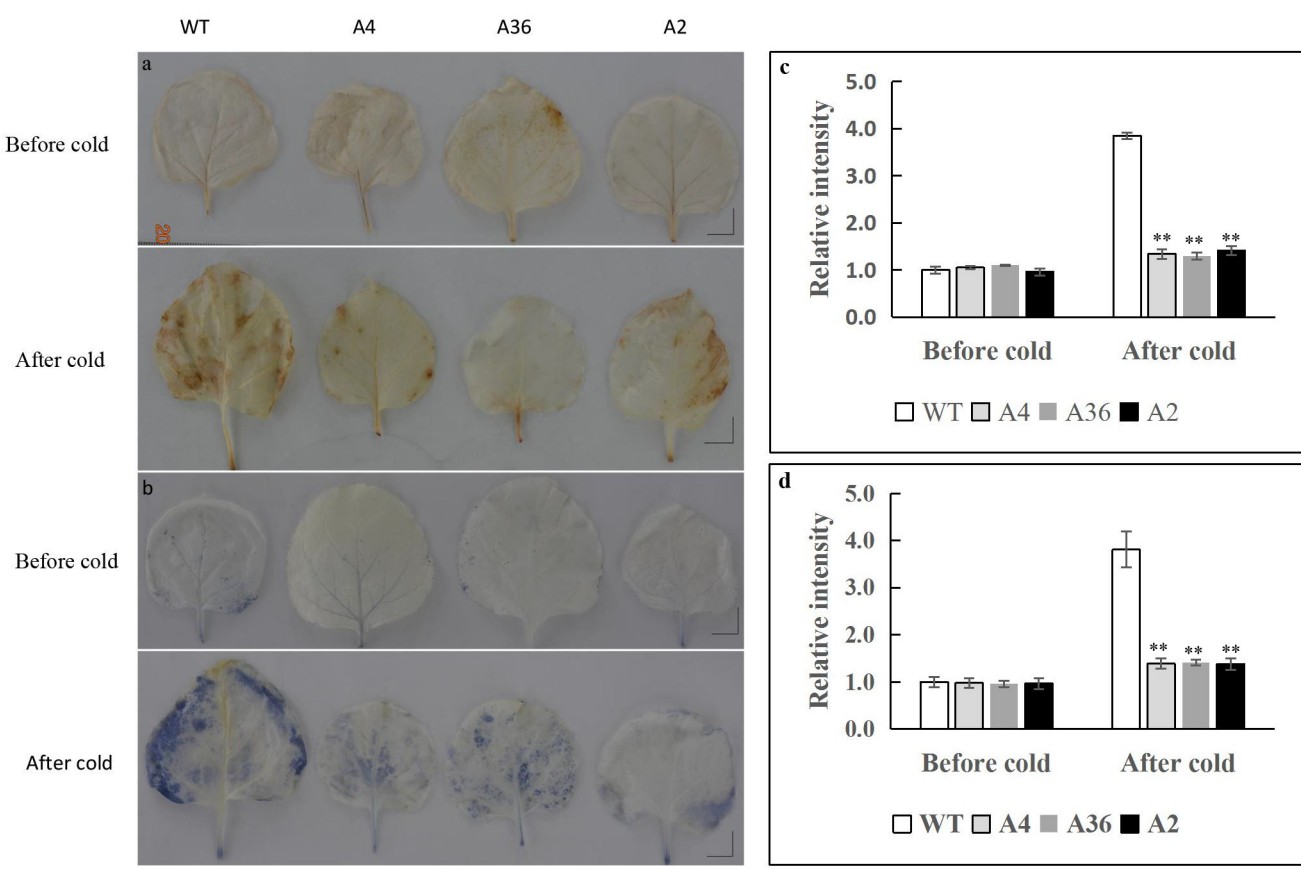

**Fig 6. The accumulation of reactive oxygen species (ROS) in WT and *MdCPK1a*-OX plants by histochemical staining.** (a) Representative photographs show in situ accumulation of $H_2O_2$ in the leaves before (upper panel) and after the cold treatment via nitro blue tetrazolium (NBT) staining. (b) in situ accumulation of $O_2^{2-}$ in WT and *MdCPK1a*-OX plants before and after the cold stress by 3,3'-diaminobenzidine (DAB). (c) Evaluation of DAB staining in the leaves of plants before and after cold stress. (d) Evaluation of NBT staining in the leaves of plants before and after cold stress. The relative staining intensities were calculated based on the staining intensity of WT plants. Error bars indicate the SEM (n = 4); *P < 0.05.

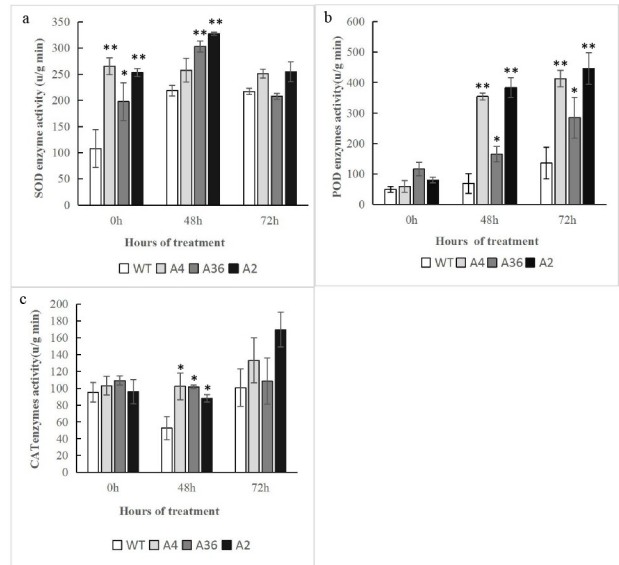

**Fig 7. Analysis of antioxidant enzyme activities in the WT and *MdCPK1a*-OX plants before and after cold treatment.** (a–c) Activities of SOD, POD and CAT, respectively. Data represent the means ± SE of at least three replicates. The significant differences between WT and *MdCPK1a*-OX plants are indicated by asterisks (*p< 0.05, **p < 0.01).

WT and the transgenic plants before and after cold treatments. The gene transcriptional levels of *NtSOD, NtGPX* and *NtCAT* were remarkably higher in transgenic tobacco whether under normal condition or cold stress, except *NtrbohD* which was lower under normal condition and significantly lower after cold stress in the transgenic tobacco. The mRNA levels of cold-responsive genes (*NtLEA5, NtSPS, NtDREB3*, except *NtERD10C*) were higher in the transgenic tobacco plants than those of WT plants under normal and cold stress condition. The expression of *NtERD10C* in the transgenic tobacco plants was similar with that in WT under normal condition, but higher under cold stress (Fig 8).

## Discussion

Calcium-dependent protein kinases respond to abiotic stress and play important roles in calcium signaling pathways. In apple, CDPKs are encoded by a multigene family consisting of 37 genes [9], however, the biological functions of which mostly remain unclear. In this study, *MdCPK1a* was identified in apple and characterized in transgenic tobacco. The sequence alignment of *MdCPK1a* with different plant CDPKs shows high similarity with stress-responsive CDPK genes such as *AtCPK1* [47, 49], *OsCDPK7* [46] and *ZmCPK1* [50]. It suggests that *MdCPK1a* might be involved in stress tolerance.

### Apple *MdCPK1a* protein localization

CDPK function is dependent on specific subcellular localization. Previous research has shown that CDPK proteins are found in cytoplasm, nucleus, the plasma membrane, oil bodies, mitochondrial outer membrane, peroxisome, and endoplasmic reticulum [6], suggesting their different functions. The N-terminal domain of CDPKs is important to subcellular localization. It is reported that membrane association is mediated by N-terminal acylation. The membrane localized CDPKs harbour a predicted N-myristoylation site and cysteine residues which would allow further palmtoylation in their N-terminus [51]. A recent study

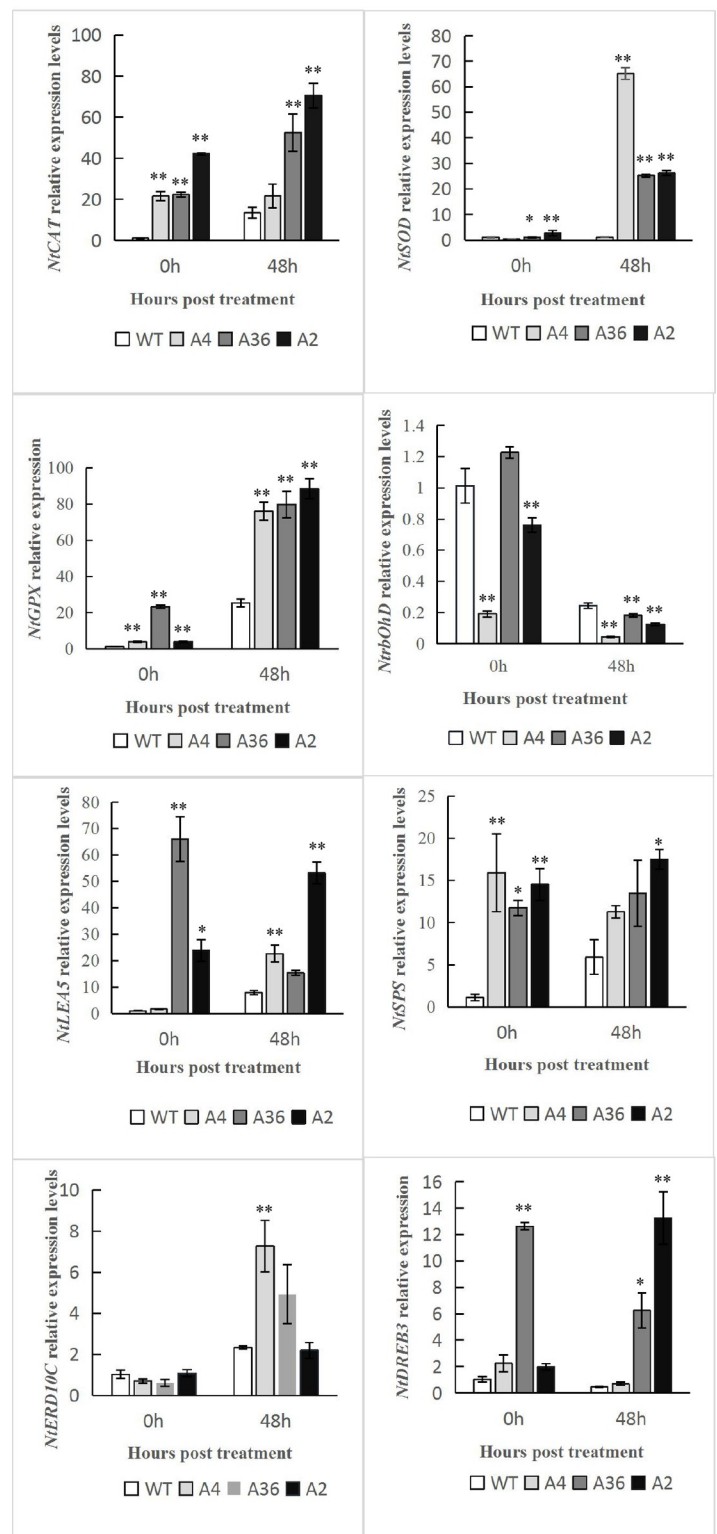

**Fig 8. The expression of the ROS-related and cold-responsive genes in WT and *MdCPK1a*-OX plants.** Data represent the means ± SE of at least three replicates. The significant differences between the WT and *MdCPK1a*-OX plants are indicated by asterisks (*p< 0.05, **p < 0.01).

revealed that *OsCPK17* has five alternative splicing (AS) forms with different subcellular localization [52]. In our experiment, MdCPK1a protein has been proved to be localized in the plasma membrane and nucleus, which is conformity with the prediction that MdCPK1a is putatively myristoylated and palmitoylated at its N-terminal (Fig 1). It indicates the post-translational modifications might allow targeting MdCPK1a protein to the plasma membrane. The location of MdCPK1a protein suggests it might participate in early signaling pathways of environmental stress by phosphorylation and activation of downstream genes [53]. The plasma membrane- or nucleus- localized CDPKs involved in abiotic stress response were also reported in other plants, such as *ZoCDPK1* in *Zingiber officinale* [54], *SiCDPK24* [29] in *Setaria italica* response to drought stress. Our results further indicate that CDPKs might have multiple subcellular localizations and involved in multiple signal transduction pathways.

### Apple *MdCPK1a* involves in response to abiotic stresses

To further understand *MdCPK1a* function, the $T_2$ plants of *MdCPK1a* overexpressing tobacco were used to study their responses to abiotic stresses. After 200 mM NaCl treatment, transgenic tobacco A4 and A36 lines showed more tolerant to salt stress, while the dry weight of shoots and roots of the transgenic lines A2 was comparable to that of the wild type, indicating that the tolerance of transgenic lines to salt stress is positive correlated to the ectopic expression levels of *MdCPK1a* in tobacco (Figs 3 and S2).

One of early responses to low temperature or other abiotic stresses in plant cell is a transient increase in cytosolic $Ca^{2+}$ derived from influx from the apoplastic space and release from internal stores [55]. $Ca^{2+}$ binding proteins can sense the transient increases of cytosolic $Ca^{2+}$, and then transmit signals to its target protein. CDPKs are the main responders in combining calcium signal with particular protein phosphorylation cascades. Although some studies showed that several *CDPK* mRNA are responsive to cold stress [8, 56–58], only a few of them were made further functional identification. In plants, as far as we know, *OsCPK7*, *OsCPK13*, *OsCPK17* and *OsCPK24* in rice and *AtCPK1* in *Arabidopsis* have been reported that participated in the response to cold stress [46, 59–62]. Furthermore, the transgenic *Arabidopsis* plants overexpressing *VaCPK20*, a *CDPK* from *Vitis amurensis* and *PeCPK10* from *Populus euphratica* improved freezing resistance [63, 64], while in *Zea mays*, *ZmCPK1* negatively regulate cold tolerance [33]. In our research, ectopic expression of *MdCPK1a* improved tobacco cold tolerance and also exhibit slightly increased salt tolerance, but no obvious improvement of drought tolerance. The cold-responsive genes, such as *NtDREB3*(dehydration-responsive element binding protein), *NtERD10C* (early response to dehydration 10C), *NtLEA5* (late embryogenesis abundant protein) and *NtSPS* (Suc-P synthase) were significantly up-regulated in the overexpressing *MdCPK1a* transgenic tobacco plants compared with the WT plants. It is known that DREBs are important transcription factors by regulating the expression of stress-responsive genes, including *ERD10C*, *LEA* and *SPS*, and so on [65, 66]. Overexpression of *MdCPK1a* increased cold-responsive genes in tobacco suggest that *MdCPK1a* may function upstream of DREBs as a positive regulator participated in the response to cold stress. Additionally, the root length of transgenic plants A4 and A36 is longer than that of WT when the seedlings of them cultured at 25˚C for 10 d on MS medium (Fig 5A). The aerial part of WT plants is little lower than that of the transgenic plants under normal condition (Fig 4). We speculated that *MdCPK1a* might also participate in the regulation of plant development. Our previous research showed that *MdCPK1a* was also induced by biotic stress [9]. These results suggest that apple *MdCPK1a* like CDPK genes in other species has overlapping functions [33, 67, 68].

## Overexpressing *MdCPK1a* enhanced tolerance to cold stress in the transgenic tobacco by scavenging ROS

Plants produced less ROS in organelles under optimal growth conditions, but under abiotic stress, the rate of ROS production is significantly elevated. ROS was produced by two major sources under abiotic stress: one is as a consequence of disruptions in metabolic activity and another is as signaling ROS which produced by NADPH oxidase [69]. ROS accumulation is a double-edged sword for plants response to abiotic stress: on the one hand, they are signaling molecules of the abiotic stress–response signal transduction network [70], on the other hand, they are also toxic byproducts that can cause oxidative destruction of cell [48, 71]. In general, CDPKs seem to function as positive regulators of ROS production in biotic stress signaling [49, 72–75], while some researches showed that CDPKs decrease ROS accumulation in abiotic stress by increasing the expression of ROS scavenging enzymes such as ascorbate peroxidase (APX), superoxide dismutase(SOD), catalase(CAT), and glutathione peroxidase(GPX) [76, 77]. For example, overexpression of the constitutively active form of oilseed rape *BnaCPK2* induces ROS accumulation and cell death through interacting with NADPH oxidase-like respiratory burst oxidase homolog D (RbohD) [78]. However, overexpression *OsCPK12* decreases ROS accumulation by increasing the expression of *OsAPx2*, *OsAPx8* and *OsrbohI* and confers increased tolerance to salt stress in rice [25]. Overexpression of *OsCPK4* in rice confers salt and drought tolerance by preventing cellular membranes from stress-induced oxidative damage [26]. In this study, overexpression of *MdCPK1a* in tobacco promoted the tolerance to cold stress by decreasing the expression of *NtrbohD* and increasing the expression of *NtSOD*, *NtCAT* and *NtGPX*. Compared with WT plants, the enzyme activities of CAT, POD, and SOD is higher and the accumulation of ROS was less in transgenic tobacco plants under cold stress. Collectively, these results suggest that *MdCPK1a* plays roles in abiotic stresses, and ectopic expression of *MdCPK1a* gene in tobacco enhances the tolerance to cold stress, which contributes to increasing the transcription levels of stress-relative genes and regulating the expression of *APX*, *CAT*, *SOD* and *rbohD* to reduce the damage to plants caused by ROS accumulation.

## Conclusion

In this research, a CDPK gene *MdCPK1a* from apple was characterized. *MdCPK1a* protein was found to localize the plasma membrane and the nucleus. Overexpression *MdCPK1a* in tobacco plants showed significantly improved their cold and salt stress tolerance than the wild type. Furthermore, Tobacco plants transfected with *MdCPK1a* showed increased resistance to cold stress by scavenging ROS accumulation and modulating the expression of stress-related genes. These results will be useful to further explore the function of *MdCPK1a* in apple.

## Supporting information

**S1 Table. Primer sequences used for cloning, subcellular localization, vector construction, transgenic confirmation and expression analysis.**
(TIF)

**S1 Fig. Phylogenetic relationship between *MdCPK1a* and other CDPK proteins.** The unrooted tree was generated using MEGA 6.0 program (http://www.megasoftware.net/) by the neighbor-joining method. Bootstrap supports from 500 replicates are indicated at each branch.
(TIF)

**S2 Fig. The identification of transgenic tobacco.** (a) Schematic representations of the vector constructs of pYH455-*MdCPK1a*. (b) Ten T1 lines of transgenic tobacco were confirmed by PCR with specific primers (PST1). (c) Six T1 lines of transgenic tobacco were confirmed by RT-PCR with specific primers (PST1). (d) Three T2 lines of transgenic tobacco were confirmed by RT-PCR with specific primers. (e) Quantification the expression of *MdCPK1a* mRNAs in the transgenic tobacco plants performed by real-time RT-PCR. RNA was extracted from the leaves of WT and *MdCPK1a*-transformed tobacco plant lines (A4, A36, and A2). Transcript abundance was normalized against the *Nttubulin* gene expression level. Data represent means and standard errors of three replicates. Significant differences between the WT and transgenic plants are indicated by asterisks($^*$p< 0.05, $^{**}$p < 0.01).
(TIF)

**S1 Raw images.**
(ZIP)

## Acknowledgments

We are very grateful to the two anonymous reviewers for critical reading of the manuscript.

## Author Contributions

**Conceptualization:** Hui Dong, Menghan Wei.

**Data curation:** Hui Dong, Changguo Luo.

**Formal analysis:** Hui Dong, Changguo Luo.

**Funding acquisition:** Shenchun Qu, Sanhong Wang.

**Investigation:** Hui Dong, Changguo Luo.

**Methodology:** Hui Dong, Chao Wu, Menghan Wei, Sanhong Wang.

**Project administration:** Shenchun Qu, Sanhong Wang.

**Resources:** Sanhong Wang.

**Supervision:** Sanhong Wang.

**Validation:** Chao Wu, Shenchun Qu.

**Visualization:** Hui Dong.

**Writing – original draft:** Hui Dong.

**Writing – review & editing:** Sanhong Wang.

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
