## [Decision Letter · Decision Letter 0]

11 Sep 2019

PONE-D-19-19438

Overexpression of MdCPK1a gene, a calcium dependent protein kinase in apple,increase tobacco cold tolerance via scavenging ROS accumulation

PLOS ONE

Dear Dr. Wang,

Thank you for submitting your manuscript to PLOS ONE. After careful consideration, we feel that it has merit but does not fully meet PLOS ONE’s publication criteria as it currently stands. Therefore, we invite you to submit a revised version of the manuscript that addresses the points raised during the review process.

We'd like to ask you to carefully read the comments of the reviewers, and revise the manuscript. 

We would appreciate receiving your revised manuscript by Oct 26 2019 11:59PM. To enhance the reproducibility of your results, we recommend that if applicable you deposit your laboratory protocols in protocols.io, where a protocol can be assigned its own identifier (DOI) such that it can be cited independently in the future. For instructions see: http://journals.plos.org/plosone/s/submission-guidelines#loc-laboratory-protocols

We look forward to receiving your revised manuscript.

Kind regards,

Hidenori Sassa

Academic Editor

PLOS ONE

Journal Requirements:

2. For reproducibility reasons we would recommend that you amend your methods section to include the source and/or deposition numbers of all the plants and seeds used in your study. We would also recommend providing some minimal details regarding the protocols references in section "Physiological measurements and histochemical staining protocol details" to allow researchers that may not have access to these references to reproduce your findings.

 [This work was supported by the National Natural Science Foundation of China (31872076, 31560551).]. 

Reviewers' comments:

Reviewer's Responses to Questions

**Comments to the Author**

1. Is the manuscript technically sound, and do the data support the conclusions?

Reviewer #1: Partly

Reviewer #2: No

2. Has the statistical analysis been performed appropriately and rigorously? 

Reviewer #1: Yes

Reviewer #2: Yes

3. Have the authors made all data underlying the findings in their manuscript fully available?

Reviewer #1: Yes

Reviewer #2: Yes

4. Is the manuscript presented in an intelligible fashion and written in standard English?

Reviewer #1: Yes

Reviewer #2: No

5. Review Comments to the Author

Reviewer #1: Authors identified CPK1a from apple and characterized it. In addition, this research revealed that MdCPK1a might play important roles in the acclimation of plants to cold stress as well as salt and heat stress via activation of ROS scavenging systems. It might be important research leading to the elucidation of the roles of CDPK in crops, however, the points below should be addressed for the publication in this journal.

Authors indicated the nuclear and cytosolic localization of MdCPK1a by GFP fusion protein. But, nuclear localization should be confirmed by DAPI staining.

In the picture taken from the top (“a” on the left panel) in Fig.3, A2 plants seem to be much larger than WT plants. However, shoot and root dry weight of A2 are comparable with that of WT plants. This discrepancy between picture and graph should be explained.

In this study, the results presented in Figure 7 are essential to support the main conclusions. Authors therefore should quantify amount of ROS with more accurate way. Especially, the ways to quantify H2O2 by fluorescent dye or kit have been established.

Although the stress response phenotypes of the transgenic plants are shown in this study, the effects of MdCPK1a expression on growth of transgenic plants under non-stressed conditions are not clearly shown. The growth phenotypes of transgenic plants need to be analyzed and presented.

Reviewer #2: The manuscript by Dong et al describes the role of a calcium-dependent protein kinase from apple, MdCPK1a, in abiotic stress tolerance, using ectopic overexpression in tobacco plants. The transgenic lines appear more tolerant to salt and cold while they behave like wild-type under drought stress. The authors further studied the cold responses to link CDPK-mediated cold tolerance and ROS detoxification, by measuring ROS level, activity of detoxifying enzymes and gene expression. However, some results are not consistent.

Major comment:

1. The 3 transgenic lines should behave similarly to link the phenotype to the expression of MdCPK1a, or at least follow the overexpression level. Yet, it is often not the case. For heat stress, the higher MdCPK1a is expressed, the lower the tolerance is induced (Fig 5) and the authors claim in the abstract l.16 that “Ectopic expression of MdCPK1a in Nicotiana benthamiana increased its salt, heat and cold resistance” while in the discussion, they state l.303-304: “ectopic expression of MdCPK1a improved tobacco cold tolerance and also exhibit slightly increased salt tolerance, but no obvious improvement of heat and drought tolerance”. For cold, lines A4 and A36 already appear much bigger than WT in control conditions when grown on plates (Fig 6a), which questions whether the increased tolerance to cold is specific or just a consequence of initial bigger size. Moreover, the analyzed molecular parameters, i.e. enzyme activity and gene expression, are not consistent in the 3 transgenic lines, which makes it difficult to link those responses to MdCPK1a overexpression.

2. In Fig 2, the fluorescent signal of MdCPK1a-GFP being very weak, it is difficult to conclude that MdCPK1a localizes to plasma membrane and not cytosol. The authors should improve the quality of the pictures and check by western-blot that the signal corresponds to the fusion protein and not to GFP alone.

3. The literature is not always relevant. For example, l. 33 ref 52 is more relevant than ref 1. L. 45, ref 6 and 10 are not relevant here. L. 263, ref 51 doesn’t deal with AtCPK1. Instead, the authors should cite ref 48 and Gao et al Plos Pathogen 2013 vol 9: e1003127. And Ref 52 is not relevant there. L. 298, the authors should include OsCPK24 (Liu et al 2018 Journal of Integrative Plant Biology vol 2 p.173-188). L. 326, ref 74 is not relevant. Instead, the authors should cite Boudsocq et al Nature 2010 vol 464 p. 418-22; Dubiella et al, PNAS 2013 vol 110 p. 8744-8749; Gao et al Plos Pathogen 2013; Kadota et al Molecular Cell 2014 vol 54, p. 43-55.

Minor comments:

1. In Fig 3a, on the left panel, line A2 seems to be as tolerant as lines A4 and A36, which is different in the right panel. Moreover, the 2 panels are not well explained in the legend. The statistics are missing in fig 3b.

2. In Fig 7, the ROS should be quantified.

3. the stress protocols are not similar in methods and results. They should be clarified.

4. the English should be improved.

5. Figure legends are inverted in Fig 6 between panels c, d, e and f l. 403-407; sup fig S1 and S2 are inverted l.427, 437.

6. PLOS authors have the option to publish the peer review history of their article (what does this mean?). If published, this will include your full peer review and any attached files.

Reviewer #1: No

Reviewer #2: No

---

## [Author Response · Author response to Decision Letter 0]

21 Oct 2019

Response to the editors: 

1. When submitting your revision, we need you to address these additional requirements. Please ensure that your manuscript meets PLOS ONE's style requirements, including those for file naming. 

 -------We changed the revised manuscript style according to PLOS ONE’s style requirements.

2. For reproducibility reasons we would recommend that you amend your methods section to include the source and/or deposition numbers of all the plants and seeds used in your study. We would also recommend providing some minimal details regarding the protocols references in section "Physiological measurements and histochemical staining protocol details" to allow researchers that may not have access to these references to reproduce your findings.

 --------In the revised manuscript, we provided more details in the section “Physiological measurements and histochemical staining”

3. PLOS ONE now requires that authors provide the original uncropped and unadjusted images underlying all blot or gel results reported in a submission’s figures or Supporting Information files. 

 -------- We have provided raw blot/gel images in supporting information.

4. Thank you for stating in your Funding Statement: [This work was supported by the National Natural Science Foundation of China (31872076, 31560551).]. 

* Please provide an amended statement that declares *all* the funding or sources of support (whether external or internal to your organization) received during this study, as detailed online in our guide for authors at http://journals.plos.org/plosone/s/submit-now. Please also include the statement “There was no additional external funding received for this study.” in your updated Funding Statement. Please include your amended Funding Statement within your cover letter. We will change the online submission form on your behalf.

------- We have provided all the funding. According the instruction, we included “There was no additional external funding receives for this study” in our updated funding statement. 

Reviewer #1: 

1. Authors indicated the nuclear and cytosolic localization of MdCPK1a by GFP fusion protein. But, nuclear localization should be confirmed by DAPI staining.

--------- Thank you for the reviewer’s suggestion! There is no doubt that the nuclear localization confirmed by DAPI staining can further guarantee the credibility of our result. We repeated the subcellular localization experiment for several times and the same results were obtained. We regretted that we did not take the photos of the DAPI staining. However, DAPI staining is not necessary for confirming nuclear localization. For example, a recent study revealed that some alternative splicing (AS) forms of OsCPK17 located in cytoplasm and nucleus. In this report, they constructed GFP-fused OsCPK17 AS isoform vectors (GFP-OsCPK17.1/2, GFP-OsCPK17.3, GFP-OsCPK17.4, and GFP-OsCPK17.5 or otherwise free GFP as a control). They used particle bombardment to identify OsCPK17 AS isoform subcellular localization in onion epidermal cells, and showed the subcellullar localization results in the Figure 3, in which they presented fluorescence projection, differential interference contrast (DIC) and merge images, but lack of DAPI staining or other nuclear markers (Almadanim et al, 2018). The similar case can also be found in the article Li et al.(2009), Plant Cell,21(2):429-441, including its correction at January 01, (2018). Plant Physiology, 177(3):1339-1341. In other case, to confirm the localization of some proteins on small organelles, such as on Golgi , Microsome, et al. It requires the organelle-markers (Almadanim et al, 2018). 

Reference: 

(1) Almadanim, M. C., Goncalves, N. M., Rosa, M. T., Alexandre, B. M., Cordeiro, A. M., Rodrigues, M., ... & Abreu, I. A. (2018). The rice cold-responsive calcium-dependent protein kinase OsCPK17 is regulated by alternative splicing and post-translational modifications. Biochimica et Biophysica Acta (BBA)-Molecular Cell Research, 1865(2), 231-246.

(2) Li, S., Lauri, A., Ziemann, M., Busch, A., Bhave, M., & Zachgo, S. (2009). Nuclear activity of ROXY1, a glutaredoxin interacting with TGA factors, is required for petal development in Arabidopsis thaliana. The Plant Cell, 21(2), 429-441.

(3) Plant Physiology Jul 2018, 177 (3) 1339-1341; DOI: 10.1104/pp.18.0066

2. In the picture taken from the top (“a” on the left panel) in Fig.3, A2 plants seem to be much larger than WT plants. However, shoot and root dry weight of A2 are comparable with that of WT plants. This discrepancy between picture and graph should be explained.

------Thank for your careful reading. After salt stress, A2 plants seem to be much larger than WT plants,while the shoot and dry weight of A2 are comparable with that of WT plants. It may be that WT plants displayed yellowing and wilted leaves under salt stress, while the leaves of transgenic plants were green and leaves fully extended, so 3 transgenic lines all seem to be larger than WT plants from the overall looking, but for the individual plants, the plant high and root size between WT and A2 plants have no significant difference, which can be seen in Fig.3(“a” on the right panel).

3. In this study, the results presented in Figure 7 are essential to support the main conclusions. Authors therefore should quantify amount of ROS with more accurate way. Especially, the ways to quantify H2O2 by fluorescent dye or kit have been established.

------ We agreed the reviewer’s comment. If we measured the content of ROS, it can provide the direct evidence, but now, for lack of the growing transgenic tobacco plants, it is difficult for us to supplement the experiment in the short time. Actually, based on the histochemical staining to determine the level of the accumulation of H2O2 and O2- has been used in many papers, such as Shen et al (2016), Trevisan et al.(2019) and Sahoo et al.(2019). For better describing the degree of staining, we supplement the column diagram of the percentage of staining area according to the statistic results of five staining leaves for each treatment in Figure 7 of the revised manuscript, which could quantify the amount of ROS indirectly. 

Figure 7

Reference:

Shen, L., Yang, S., Yang, T., Liang, J., Cheng, W., Wen, J., ... & Shi, W. (2016). CaCDPK15 positively regulates pepper responses to Ralstonia solanacearum inoculation and forms a positive-feedback loop with CaWRKY40 to amplify defense signaling. Scientific reports, 6, 22439.

Trevisan, Sara, et al. Nitrate affects transcriptional regulation of UPBEAT1 and ROS localisation in roots of Zea mays L. Physiologia plantarum 166.3 (2019): 794-811.

Sahoo, S., Saha, B., Awasthi, J. P., Omisun, T., Borgohain, P., Hussain, S., ... & Panda, S. K. (2019). Physiological introspection into differential drought tolerance in rice cultivars of North East India. Acta Physiologiae Plantarum, 41(4), 53.

4. Although the stress response phenotypes of the transgenic plants are shown in this study, the effects of MdCPK1a expression on growth of transgenic plants under non-stressed conditions are not clearly shown. The growth phenotypes of transgenic plants need to be analyzed and presented.

--------The growth phenotypes of transgenic plants has been presented in this paper actually. The Fig 6a shows the phenotype of WT, A4, A36 and A2 at the normal growth temperature for 10 days on MS medium after germinated. It showed that the root length of transgenic plants A4 and A36 is longer, but only of A4 is significant longer than that of WT. In the panel of “before drought” of Figure 4 , the growth of transgenic plants under the normal conditions are also shown. It shows that the aerial part of WT plants is little higher than that of the transgenic plants under normal condition. We also speculated that MdCPK1a might participate in the regulation of plant development.

Thus，in the discussion of revised manuscript, in line 449-454, we added the discussion as below: “Additionally, the root length of transgenic plants A4 and A36 is longer than that of WT when the seedlings of them cultured at 25℃ for 10 d on MS medium( Fig 6a) . The aerial part of WT plants is a little higher than that of the transgenic plants under normal condition (Fig 4). We speculated that MdCPK1a might also participate in the regulation of plant development.”

Reviewer #2: 

Major comment:

1. The 3 transgenic lines should behave similarly to link the phenotype to the expression of MdCPK1a, or at least follow the overexpression level. Yet, it is often not the case. For heat stress, the higher MdCPK1a is expressed, the lower the tolerance is induced (Fig 5) and the authors claim in the abstract l.16 that “Ectopic expression of MdCPK1a in Nicotiana benthamiana increased its salt, heat and cold resistance” while in the discussion, they state l.303-304: “ectopic expression of MdCPK1a improved tobacco cold tolerance and also exhibit slightly increased salt tolerance, but no obvious improvement of heat and drought tolerance”. For cold, lines A4 and A36 already appear much bigger than WT in control conditions when grown on plates (Fig 6a), which questions whether the increased tolerance to cold is specific or just a consequence of initial bigger size. Moreover, the analyzed molecular parameters, i.e. enzyme activity and gene expression, are not consistent in the 3 transgenic lines, which makes it difficult to link those responses to MdCPK1a overexpression.

--------We apologized for confusing the reader because of our carelessness. Ectopic expression of MdCPK1a improved tobacco cold and salt tolerance, but no obvious improvement of heat and drought tolerance.The description in the summary is incorrect and has been modified. 

AS for the comment “whether the increased tolerance to cold is specific or just a consequence of initial bigger size?” CDPK regulating development in plant has been reported in some studies ( see review by Singh et al. 2017 ). In our study, we also supposed that MdCPK1a might involve in plant development (Fig 6a). We added the discussion in L449-L454. On the other hand, we don’t think that the bigger size of transgenic plants will influence on the cold resistance ability in this experiment. We think the cold stress related parameters, such as the content of MDA，the electrolyte leakage，the enzyme activities, et al. do not change with the plant size. 

In our research, it was no doubt that the stress related enzyme activities and the stress related gene expression of the transgenic plants were higher that those of control plant. The transgenic tobacco plants are more resistant to cold and salt stresses as result of overexpression of MdCPK1a. Why are the stress related enzyme activities and gene expression not consistent with the gene expression of MdCPK1a ? We explained that the posttranscriptional or posttranslational modification of MdCPK1a might exist in MdCPK1a -OX plants, which might be influenced by the insertion locus of MdCPK1a. 

Reference:

Singh, A., Sagar, S., & Biswas, D. K. (2017). Calcium dependent protein kinase, a versatile player in plant stress management and development. Critical reviews in plant sciences, 36(5-6), 336-352.

2. In Fig 2, the fluorescent signal of MdCPK1a-GFP being very weak, it is difficult to conclude that MdCPK1a localizes to plasma membrane and not cytosol. The authors should improve the quality of the pictures and check by western-blot that the signal corresponds to the fusion protein and not to GFP alone.

 ------- Indeed, the fluorescent signal of MdCPK1a-GFP is weak on plasma membrane. We also believe it will make our conclusion more convincing if we conducted western-blot. The similar comment was given on the subcelluar localization by Reviewer #1. Please refer to our response to the comment to Reviewer #1. According to the reviewer’s suggestion, we improved the quality of the image. It became more clearly.

3. The literature is not always relevant. 

------Thank you very much for careful reading! According the reviewer’s suggestion, we corrected the references accordingly.

For example, l. 33 ref 52 is more relevant than ref 1. 

------We corrected it.

L. 45, ref 6 and 10 are not relevant here.

------We deleted the two reference.

 L. 263, ref 51 doesn’t deal with AtCPK1. Instead, the authors should cite ref 48 and Gao et al Plos Pathogen 2013 vol 9: e1003127. And Ref 52 is not relevant there. 

-------We corrected it.

L. 298, the authors should include OsCPK24 (Liu et al 2018 Journal of Integrative Plant Biology vol 2 p.173-188). 

-------We added OsCPK24 in the manuscript, and cited the reference by Liu et al (2018).

L. 326, ref 74 is not relevant. Instead, the authors should cite Boudsocq et al Nature 2010 vol 464 p. 418-22; Dubiella et al, PNAS 2013 vol 110 p. 8744-8749; Gao et al Plos Pathogen 2013; Kadota et al Molecular Cell 2014 vol 54, p. 43-55.

-------We corrected it.

Minor comments:

1. In Fig 3a, on the left panel, line A2 seems to be as tolerant as lines A4 and A36, which is different in the right panel. Moreover, the 2 panels are not well explained in the legend. The statistics are missing in fig 3b.

-------The comment was also given by the Reviewer #1. Please refer to our response above. Thank you for your careful reviewing! The shoot dry weight of A4 and A36 (except A2) was significantly higher than that of WT; The root dry weight of A4 (except A36 and A2)was significantly higher than that of WT. We added the statistics on the Fig.3b.

2. In Fig 7, the ROS should be quantified.

-------We responded the same comment to the Reviewer #1 on the above.

3. the stress protocols are not similar in methods and results. They should be clarified.

-------We read carefully and already kept the stress protocol of methods and results consistent.

4. the English should be improved.

------- We polished the English.

5. Figure legends are inverted in Fig 6 between panels c, d, e and f l. 403-407; sup fig S1 and S2 are inverted l.427, 437.

------- Thank you so much for your careful reading! We corrected it .

---

## [Decision Letter · Decision Letter 1]

12 Dec 2019

PONE-D-19-19438R1

Overexpression of MdCPK1a gene, a calcium dependent protein kinase in apple,increased tobacco cold tolerance via scavenging ROS accumulation

PLOS ONE

Dear Prof. Wang,

Thank you for submitting your manuscript to PLOS ONE. After careful consideration, we feel that it has merit but does not fully meet PLOS ONE’s publication criteria as it currently stands. Therefore, we invite you to submit a revised version of the manuscript that addresses the points raised during the review process.

Please carefully read the reviewers' comments and revise the manuscript. Especially, the Reviewer 2's comments are critical.

We would appreciate receiving your revised manuscript by Jan 26 2020 11:59PM. To enhance the reproducibility of your results, we recommend that if applicable you deposit your laboratory protocols in protocols.io, where a protocol can be assigned its own identifier (DOI) such that it can be cited independently in the future. For instructions see: http://journals.plos.org/plosone/s/submission-guidelines#loc-laboratory-protocols

We look forward to receiving your revised manuscript.

Kind regards,

Hidenori Sassa

Academic Editor

PLOS ONE

Reviewers' comments:

Reviewer's Responses to Questions

**Comments to the Author**

1. If the authors have adequately addressed your comments raised in a previous round of review and you feel that this manuscript is now acceptable for publication, you may indicate that here to bypass the “Comments to the Author” section, enter your conflict of interest statement in the “Confidential to Editor” section, and submit your "Accept" recommendation.

Reviewer #1: All comments have been addressed

Reviewer #2: (No Response)

2. Is the manuscript technically sound, and do the data support the conclusions?

Reviewer #1: (No Response)

Reviewer #2: Partly

3. Has the statistical analysis been performed appropriately and rigorously? 

Reviewer #1: Yes

Reviewer #2: Yes

4. Have the authors made all data underlying the findings in their manuscript fully available?

Reviewer #1: Yes

Reviewer #2: No

5. Is the manuscript presented in an intelligible fashion and written in standard English?

Reviewer #1: Yes

Reviewer #2: No

6. Review Comments to the Author

Reviewer #1: Basically, authors properly answered the comments by explaining the detail. However, I found another point to be modified. Fig.4 might be important pictures that also show difference in growth even under normal condition. Scale bars should be indicated to further provide information of plant size.

Reviewer #2: The authors answered part of the initial comments but some points still need clarification.

Major comments:

1. The part on heat tolerance is confusing since the authors conclude l. 434-436 that “ectopic expression of MdCPK1a exhibit…no obvious improvement of heat and drought tolerance” while in the results, they state l.310-311 “the survival rates of…A36 (24%) and A2 (98%) were remarkably higher than WT”. Indeed, the thermotolerance is inversely correlated with MdCPK1 expression level in the transgenics, which makes it difficult to conclude without any additional investigation. Thus, the whole part on heat stress tolerance should be deleted from this manuscript.

2. The picture of MdCPK1 localization has been more contrasted. However, the membrane localization is still not demonstrated. Indeed, some cytoplasmic strands are visible and MdCPK1 could be located in cytosol and/or plasma membrane. This experiment is important to correlate with the acylation prediction of MdCPK1. Either the authors balance their statement, or they provide additional data such as co-localization with known plasma membrane protein, or western-blot of proteins extracted in the presence/absence of triton X100 which will extract total proteins including the membrane ones (presence) or only soluble ones (absence). Moreover, a western-blot will additionally prove that the fluorescence observed is due to MdCPK1-GFP fusion and not GFP alone. Finally, “plasma membrane” should be deleted l.265-266 since free GFP doesn’t go to membrane. It only diffuses to cytosol and nucleus.

Minor comments:

1. In Fig3, the discrepancy between pictures (a, left panel) and graphs (b, left panel) is surprising. Maybe measuring fresh weight would have been more relevant here. Nevertheless, the authors could comment on that: it suggests that under salt stress, MdCPK1 transgenics retain water better than WT.

2. In Fig7, the authors added quantification of the ROS staining. The corresponding legend is missing l.358-362, the panels c and d should be cited in the results l.352, and the way they quantified the staining should be explained in the methods.

3. In the discussion l.447-448, the authors state that “the aerial part of WT plants is little higher than that of transgenics under normal conditions”. However, the picture in Fig4 shows that WT plants are actually smaller than the transgenics in control conditions. This should be corrected.

4. The stress protocols are still different in methods and figures. In results, salt stress (l.285) and drought (l.303) are performed for 25d on 4-week-old plants while in the methods, 4-week-old plants are further grown for 14d before applying stress, ie 6-week-old before stress (l.169-170). Also correct accordingly if needed the legend l.305-307.

5. L.86, Ralstonia solanac should be Ralstonia solanacearum. L.402, predicated should be predicted.

6. English still needs some improvement, especially l.89-91, l.113-115, l.178-180.

7. L.251, the sentence should be rephrased by “the selected CDPK proteins were clustered into three subgroups” because the authors just didn’t include CDPKs from subgroup IV. L.252, ZmCDPK1 should be “ZmCPK1”.

8. In FigS2, the qRT-PCR (panel e) is missing.

9. Fig6 has been reorganized but the labels are now wrong: a and b are inverted, c and d, and e and f, as well. Moreover, the genotype labelling is missing in panel b (which should be “a” with MS and MS+4°C).

10. L.375-378, the sentence should be modified because NtSPS and NtLEA5 are already strongly induced in MdCPK1 transgenics in normal conditions.

7. PLOS authors have the option to publish the peer review history of their article (what does this mean?). If published, this will include your full peer review and any attached files.

Reviewer #1: No

Reviewer #2: No

---

## [Author Response · Author response to Decision Letter 1]

28 Sep 2020

Thank you for the two reviewers and the editors

 We are grateful to your critical reading. Your comments are invaluable for the manuscript conforming to the requirement for publishing. The comments and responses were listed as follows. We hope the revised manuscript will meet the requirements for publication on Plos One.

Reviewer #1: Basically, authors properly answered the comments by explaining the detail. However, I found another point to be modified. Fig.4 might be important pictures that also show difference in growth even under normal condition. Scale bars should be indicated to further provide information of plant size.

Reply：Thank you for the reviewer’s suggestion！We have indicated the scale bars in the figure 4 and the legends. Bars =10 cm.

Reviewer #2: The authors answered part of the initial comments but some points still need clarification.

Major comments:

1.The part on heat tolerance is confusing since the authors conclude l. 434-436 that “ectopic expression of MdCPK1a exhibit…no obvious improvement of heat and drought tolerance” while in the results, they state l.310-311 “the survival rates of…A36 (24%) and A2 (98%) were remarkably higher than WT”. Indeed, the thermotolerance is inversely correlated with MdCPK1 expression level in the transgenics, which makes it difficult to conclude without any additional investigation. Thus, the whole part on heat stress tolerance should be deleted from this manuscript.

Reply: Thank you for the reviewer’s suggestion. We deleted the whole part of heat stress in the manuscript.

2. The picture of MdCPK1 localization has been more contrasted. However, the membrane localization is still not demonstrated. Indeed, some cytoplasmic strands are visible and MdCPK1 could be located in cytosol and/or plasma membrane. This experiment is important to correlate with the acylation prediction of MdCPK1. Either the authors balance their statement, or they provide additional data such as co-localization with known plasma membrane protein, or western-blot of proteins extracted in the presence/absence of triton X100 which will extract total proteins including the membrane ones (presence) or only soluble ones (absence). Moreover, a western-blot will additionally prove that the fluorescence observed is due to MdCPK1-GFP fusion and not GFP alone. Finally, “plasma membrane” should be deleted l.265-266 since free GFP doesn’t go to membrane. It only diffuses to cytosol and nucleus.

Reply: Thank you for your suggestion. We conducted the supplementary experiment of Western Blot to further confirm the subcellular localization of MdCPK1 according to the reviewer’ suggestion. 

The Western blot assay showed that MdCPK1a-GFP was exclusively detected in the cell membrane and nucleus fractions, but not in the cytosolic by immunoblotting with anti-GFP antibody (Figure 2c). The results indicated that MdCPK1a protein was localized to the nucleus and cell membrane. The result was added in the revised manuscript and Figure 2 was modified also accordingly..

Minor comments:

1. In Fig3, the discrepancy between pictures (a, left panel) and graphs (b, left panel) is surprising. Maybe measuring fresh weight would have been more relevant here. Nevertheless, the authors could comment on that: it suggests that under salt stress, MdCPK1 transgenics retain water better than WT.

Reply: we agreed with the reviewer’s comments that fresh weight would have been more relevant here. However, we believed that dry weight might more accurately reflect difference of the growth and development of lines. A transgenic line will increase more biomass and accumulate more dry matter if it has higher adaptability under certain condition, accordingly, the dry weight of the plant is higher than the others. We don’t think the picture a(left) and graph of b(left) is discrepant. 

2. In Fig7, the authors added quantification of the ROS staining. The corresponding legend is missing l.358-362, the panels c and d should be cited in the results l.352, and the way they quantified the staining should be explained in the methods.

Reply: We apologized for our carelessness. The way of quantification for the staining have been added in the methods. The corresponding legend has been added for the Fig7 (in the revised manuscript, it was renamed Fig.6) and the panels c and d were cited in the results.

3. In the discussion l.447-448, the authors state that “the aerial part of WT plants is little higher than that of transgenics under normal conditions”. However, the picture in Fig4 shows that WT plants are actually smaller than the transgenics in control conditions. This should be corrected.

Reply: We apologized for our carelessness, we have corrected the mistake.

4. The stress protocols are still different in methods and figures. In results, salt stress (l.285) and drought (l.303) are performed for 25d on 4-week-old plants while in the methods, 4-week-old plants are further grown for 14 d before applying stress, ie 6-week-old before stress (l.169-170). Also correct accordingly if needed the legend l.305-307.

Reply: Thank you for the reviewer’s careful reading. We corrected the mistyping. The transgenic lines were all treated at 6 weeks old.

5. L.86, Ralstonia solanac should be Ralstonia solanacearum. L.402, predicated should be predicted.

Reply: Thank so much for your comments! we have corrected the error.

6. English still needs some improvement, especially l.89-91, l.113-115, l.178-180.

Reply: Thank you! We proofread the whole manuscript and improved the three sentences as follows:

L89-91: Conversely, some CDPKs play negative regulators of stress response for transgenic plants overexpressed them show more sensitive to abiotic/biotic stresses.

L113-115: The fourth and fifth young leaves were taken from the annual branches of the Malus domestica cv. ‘Jonathan’growing in the greenhouse. Total RNA was extracted by using CTAB method.

L.178-180: Leaf samples (0.5 g) were homogenized in 2 mL 20% trichloroacetic acid with the aid of some sand, and then the homogenate was centrifuged at 16,000 g for 20 min at 4 °C. The supernatant (1 mL) was mixed with equal volume of 0.5% (w/v) TBA.

7. L.251, the sentence should be rephrased by “the selected CDPK proteins were clustered into three subgroups” because the authors just didn’t include CDPKs from subgroup IV. L.252, ZmCDPK1 should be “ZmCPK1”.

Reply: Thank you for your suggestion! We have rephrased the sentence in the revised manuscript.

8. In FigS2, the qRT-PCR (panel e) is missing.

Reply: We added the qPCR (panel e) in the new FigS2.

9. Fig6 has been reorganized but the labels are now wrong: a and b are inverted, c and d, and e and f, as well. Moreover, the genotype labelling is missing in panel b (which should be “a” with MS and MS+4°C).

Reply: Thank you! We have reorganized the Fig5 and re-labelled (the original figure 6 was revised as figure 5 in the revised manuscript).

10. L.375-378, the sentence should be modified because NtSPS and NtLEA5 are already strongly induced in MdCPK1 transgenics in normal conditions.

Reply: we have modified the sentence.

The mRNA levels of cold-responsive genes except NtERD10C were higher in the transgenic tobacco than those of WT plants under normal and cold stress condition (Fig 8). The expression of NtERD10C in the transgenic tobacco was similar with that in WT under normal condition, but higher under cold stress.

---

## [Decision Letter · Decision Letter 2]

28 Oct 2020

Overexpression of MdCPK1a gene, a calcium dependent protein kinase in apple,increased tobacco cold tolerance via scavenging ROS accumulation

PONE-D-19-19438R2

Dear Dr. Wang,

We’re pleased to inform you that your manuscript has been judged scientifically suitable for publication and will be formally accepted for publication once it meets all outstanding technical requirements.

Kind regards,

Hidenori Sassa

Academic Editor

PLOS ONE

Additional Editor Comments (optional):

Reviewers' comments:

Reviewer's Responses to Questions

**Comments to the Author**

1. If the authors have adequately addressed your comments raised in a previous round of review and you feel that this manuscript is now acceptable for publication, you may indicate that here to bypass the “Comments to the Author” section, enter your conflict of interest statement in the “Confidential to Editor” section, and submit your "Accept" recommendation.

Reviewer #1: All comments have been addressed

Reviewer #2: (No Response)

2. Is the manuscript technically sound, and do the data support the conclusions?

Reviewer #1: Yes

Reviewer #2: Yes

3. Has the statistical analysis been performed appropriately and rigorously? 

Reviewer #1: Yes

Reviewer #2: Yes

4. Have the authors made all data underlying the findings in their manuscript fully available?

Reviewer #1: Yes

Reviewer #2: Yes

5. Is the manuscript presented in an intelligible fashion and written in standard English?

Reviewer #1: Yes

Reviewer #2: No

6. Review Comments to the Author

Reviewer #1: I think authors properly answered the comments from reviewers. Indeed, the manuscript was much improved after the revisions. For me, this manuscript is currently sufficient for the publication in this journal.

Reviewer #2: The authors answered almost to all my comments. There are just few minor editing points left:

1. English still needs improvement. L. 87-88 could be “Conversely, some CDPKs are negative regulators of stress responses because transgenic plants overexpressing them are more sensitive to abiotic/biotic stresses.” L.352-354 could be “To know whether MdCPK1a regulates ROS levels in cold response, we compared the ROS levels in the overexpressing tobacco lines and WT plants after suffering cold stress.”

2. l.191, “3 ml” has been replaced by “equal volume”. But this would correspond to 1 ml, and not 3 ml. Please correct or confirm the good value.

3. l.259, and l.399, ZmCDPK1 should be ZmCPK1.

4. L.469, ref 48 should be ref 49.

7. PLOS authors have the option to publish the peer review history of their article (what does this mean?). If published, this will include your full peer review and any attached files.

Reviewer #1: No

Reviewer #2: No

---

## [Editor Report · Acceptance letter]

3 Nov 2020

PONE-D-19-19438R2 

Overexpression of *MdCPK1a* gene, a calcium dependent protein kinase in apple，increase tobacco cold tolerance via scavenging ROS accumulation 

Dear Dr. Wang:

I'm pleased to inform you that your manuscript has been deemed suitable for publication in PLOS ONE. Congratulations! Your manuscript is now with our production department. 

Kind regards, 

on behalf of

Dr. Hidenori Sassa 

Academic Editor

PLOS ONE